# Quality Control and Authentication of Argan Oils: Application of Advanced Analytical Techniques

**DOI:** 10.3390/molecules28041818

**Published:** 2023-02-15

**Authors:** Meryeme El Maouardi, Mourad Kharbach, Yahya Cherrah, Kris De Braekeleer, Abdelaziz Bouklouze, Yvan Vander Heyden

**Affiliations:** 1Biopharmaceutical and Toxicological Analysis Research Team, Laboratory of Pharmacology and Toxicology, Faculty of Medicine and Pharmacy, University Mohammed V, Rabat 10100, Morocco; 2Department of Analytical Chemistry, Applied Chemometrics and Molecular Modelling, Vrije Universiteit Brussel (VUB), Laarbeeklaan 103, 1090 Brussels, Belgium; 3Research Unit of Mathematical Sciences, University of Oulu, 90014 Oulu, Finland; 4Pharmacognosy, Bioanalysis & Drug Discovery Unit, Faculty of Pharmacy, University Libre Brussels, 1050 Brussels, Belgium

**Keywords:** argan oil, spectroscopic techniques, chromatographic techniques, quality control, authentication, chemometrics

## Abstract

In addition to the nutritional and therapeutic benefits, Argan oil is praised for its unique bio-ecological and botanic interest. It has been used for centuries to treat cardiovascular issues, diabetes, and skin infections, as well as for its anti-inflammatory and antiproliferative properties. Argan oil is widely commercialized as a result of these characteristics. However, falsifiers deliberately blend Argan oil with cheaper vegetable oils to make economic profits. This reduces the quality and might result in health issues for consumers. Analytical techniques that are rapid, precise, and accurate are employed to monitor its quality, safety, and authenticity. This review provides a comprehensive overview of studies on the quality assessment of Moroccan Argan oil using both untargeted and targeted approaches. To extract relevant information on quality and adulteration, the analytical data are coupled with chemometric techniques.

## 1. Introduction

The Argan tree (*Argania spinosa*. L), belonging to the Sapotaceae family, is a topical plant appearing endemically in southwestern Morocco (Appendix A [1]). It covers approximately an area of 870,000 ha [2] and protects soil from erosion and desertification. Because of its resistance to heat and drought, it is widespread in the arid and semi-arid southwestern regions. In the last decades, the Argan forest suffered from deforestation, estimated at 600 ha per year due to human activities [3]. As a consequence, since 1998, the UNESCO has put Argan forests on the list of reserves of the protected biosphere [4].

*Argania spinosa* is a source of valuable oil, which is not only used as human food, but also in traditional medicine and in pharmaceutical and cosmetic preparations. Several studies demonstrated its pharmacological activities, such as blood pressure reduction [5], anti-inflammatory- [6], antiproliferative- [7], cardioprotective- [7,8,9], antidiabetic- [8], antioxidant- [10] and hypolipidemic- [8] activities. The production of Argan oil (AO) is of social, environmental and economic interest [11]. Argan oil is luxurious because of its rarity, laborious preparation, nutritional and pharmacological properties [4]. Thus, it became very popular on the world markets, and nowadays is only exported by Moroccan producers. The balanced chemical composition determines the commercial value of Argan oil. The limited production, the high price and the high demand for Argan oil makes it susceptible to deliberate adulteration with vegetable oils for financial profit [12]. Adulteration of Argan oil leads to consumer’s dissatisfaction and to a product with less health benefits.

Generally, the quality of Argan oil is specified by chemical and physical properties, which deliver information on both the nutritive and sensorial quality. These properties include color, taste, flavor, peroxide index, moisture, iodine value, saponification value, metal contents, fatty acid-, tocopherol-, polyphenol-, and sterol compositions [4]. A Moroccan Normative (N.M. 08.5.090) guideline was published in 2003 to state the quality specifications of Argan oil [12]. The main objective of this guideline is to protect producers against counterfeit products and to assure consumers about the specific product. In 2009, under the impulse of the Mohammed VI Foundation for Research and Protection of the Argan Tree, Moroccan authorities have established an internal protection system leading to the Argan Protected Geographical Indication (PGI) to list the product on the National register, with the creation of the Moroccan Association of the Geographical Indication of Argan Oil (AMIGHA). The constitution of this legal framework protects the original Argan products, on the one hand, and links the concept of quality with the geographical origin on the other. Since 25 January 2010, the Argan PGI is operative, which permits assigning the label “Huile d’Argan” exclusively to products from the south west of Morocco, being prepared according to well defined procedures recorded in the book of specifications [12].

Fingerprinting techniques refer to a variety of analytical techniques (including chromatographic and spectroscopic ones) that provide analytical signals (fingerprints) of which the entire profile is used for the purpose of specification, quality evaluation, identification and authentication of foodstuffs [13].

In this context, this work provides a comprehensive review of the fingerprint analysis of Moroccan Argan oil using fingerprints collected by means of analytical techniques (e.g., Fourier Transform Mid-Infrared (FTIR), Near-Infrared (FT-NIR), Raman, Ultraviolet–Visible and Nuclear Magnetic Resonance (NMR) spectroscopy, mass spectrometry, fluorimetry, HPLC, GC, electro thermal atomization, atomic absorption spectroscopy, inductively coupled plasma atomic emission spectroscopy, electronic nose and voltammetric electronic tongue), and treated with chemometric data analysis techniques, in order to assess, for instance, its extraction-process type, geographic origin, identification, and adulteration level.

## 2. Argan Oil Extraction

Argan oil has, for many years, been produced traditionally by the female population living in the Argan forests. Nowadays, thanks to industrial advances, Argan oil is produced with high quality and in large quantity. The Argan oil can be extracted by three different approaches, i.e., traditional, mechanical and solvent extraction. Those methods are utilized to produce two types of Argan oil, edible and cosmetic, depending on whether the kernels were roasted before extraction or not (Figure 1). In laboratories, cosmetic Argan oil is directly obtained using a lipophilic solvent after pulverizing the kernels. In order to achieve the best extraction efficiency and to obtain the best chemical composition of the extra virgin oil, besides the solvents, extraction by CO_2_ under supercritical conditions is also performed [12]. The physicochemical parameters of the extracted oils obtained by supercritical fluid extraction from (SFE) and traditional methods are comparable. Therefore, SFE, as a technique used for oil processing, does not markedly alter the quality of Argan [14].

Traditionally, for centuries, edible Argan oil was prepared by Berber women. The peel and pulp of collected Argan fruits are first discarded. The Argan seeds obtained are broken carefully with a stone. The isolated kernels are air-dried, then roasted. From these latter, the oil is extracted using a traditional manual millstone. The produced Argan dough is hand mixed with a small quantity of warm water to facilitate the kneading. The wet dough obtained is then hand pressed to extract the Argan oil. The residue is called “Argan press cake”. The roasting step produces an oil with a hazelnut taste and a brown to red color. The traditionally prepared Argan oil suffers from disadvantages, including microbiological contamination due to bad water quality, which may cause a limited shelf-life stability. The taste and smell are variable and sometimes unsatisfactory for consumers. In addition, the unsatisfactory quality also reduces the storage time of this Argan oil to maximally a few weeks [4]. Moreover, the process is time consuming (ten work hours to get one liter of oil), hard, and is applied only to produce roasted Argan oil with low quality and a yield of about 30% [15].

The mechanical extraction, named also half-industrialized extraction, uses a mechanical press technique to extract the either roasted or unroasted Argan oils. This technique follows the same steps as the traditional method, except that the kernels are directly pressed, and the mixing of the dough does not need water. Therefore, the quality of the produced oil and its conservation time are better. The press cake (kernel residues) now contains less than 10% of oil [11]. In the cooperative, only half an hour is needed to obtain one liter of oil. The yield is higher compared to the traditional method, i.e., around 45% [15].

Unroasted Argan oil (cosmetic grade) can be prepared in industry or the laboratory from kernels using a volatile lipophilic solvent, generally halogenated, such as chloroform or dichloromethane [16]. To avoid the oxidation of the fatty acids a lipophilic antioxidant, such as ascorbyl palmitate, is added [17]. This process increases the yield to 50–55% and allows obtaining a stable lipidic extract without strong odor. However, solvent extracted oil from unroasted kernels has unsatisfactory organoleptic properties compared to the traditionally or press-extracted oil [15]. The oil may also contain some solvent contamination and production is very expensive compared to mechanical extraction. 

Extraction with CO_2_ under supercritical conditions has also been performed because of its extraction efficiency and to obtain the best quality of Argan oil. In 2020, research was conducted by González-Fernández et al. on a new environment-friendly technique for the extraction of AO [18]. The use of water or ecological solvents, as ethanol and ethyl acetate, has been studied in order to evaluate their impact on AO quality and composition. These ecological solvents allowed a good AO quality and may be used in the food and cosmetic industries. 

In addition to solvents, other factors, such as temperature, could increase the yield. Recently, Ouchbani et al. studied the effects of heating during the press-extraction process on the yield [7]. The highest yield values (53–55%) were obtained at 80 °C without compromising the quality of the oil.

## 3. Argan Oil Composition

The chemical composition of AO is well balanced for dietary or cosmetic use. Argan oil has also many pharmacological activities and health benefits due to its compositional profile, containing acylglycerides, fatty acids, tocopherols, unsaponifiables fractions and other minor compounds. 

The triacylglycerol (TAG) fraction constitutes the major one in Argan oil (around 99% of the saponifiable fraction) [7]. The major TAGs quantified in virgin Argan oil are di-oleoyl-linoleoyl-glycerol (OOL) (19.5%), di-linoleoyl-oleoyl-glycerol (LLO) (13.6%), palmitoyl-oleoyl-linoleoyl-glycerol (POL) (13.6%), tri-olein (OOO) (12.8%) and palmitoyl-di-oleoyl-glycerol (POO) (11.5%) [6,7,8,9,10]. 

The fatty-acid composition of Argan oil consists of long chains of saturated, unsaturated (80%) and polyunsaturated fatty acids. They are essential for health promotion through the diet [17]. The predominant unsaturated fatty acids are oleic ω-9 (43–49%) and linoleic acid ω-6 (29–36%). Other, such as arachidic, behenic, eicosenoic, heptadecanoic, linolenic, nonadecenoic, myristic, and palmitoleic acids occur in trace amounts [17]. The two major saturated fatty acids present in Argan oil are palmitic and stearic acid. The percentage of each component may differ due to different parameters, among which the extraction process [17]. Appendix A shows the percentages of fatty acids in Argan oil as reported in previous studies. Some studies determined the fatty-acid composition in a single region [19], while others, such as Kharbach et al. [20], determined and compared the amounts of fatty acids in different Moroccan regions. Despite the different geographical origin, the fatty-acid composition is within the range described by the regulations. Table 1 allows a comparison of Argan oil and other vegetable oils in terms of their fatty-acid composition. Argan oil contains a high percentage of stearic acid (4.3–7.2%), which allows differentiating it from other vegetable oils except from sunflower oil, which seems to have a similar percentage of stearic acid (2.5–7.0%).

The unsaponifiable matter forms 1% of the Argan-oil composition. It encompasses carotenes (37%), tocopherols (8%), triterpene alcohols (20%), sterols (29%) and xantophylles (5%) [11]. 

Extra virgin Argan oil is a rich source of tocopherols (600–900 mg/kg). It contains 69 mg/kg oil of α-tocopherol, 4.56 mg/kg of β-tocopherol, 802 mg/kg of γ-tocopherol, and 104 mg/kg of δ-tocopherol. β, γ and δ-tocopherols are known for their antioxidant activity, and are responsible for the oil shelf stability and conservation [25]. It has been shown that the tocopherol composition is influenced by the geographical origin of the samples. The highest tocopherol contents occur in samples from Tiznit (919.26 mg/kg) and Ait- Baha (906.21 mg/kg of oil) [20].

## 4. Argan Oil Quality Control and Authentication Assessment

Argan oil has been widely used in the traditional medicine for hundreds of years. It is incorporated in pharmaceutical and cosmetic preparations [26]. Chemical profiles (fingerprints) of Argan oil can be obtained by spectroscopic or chromatographic techniques. These fingerprints may be useful for the identification, classification and quality control of the oil. A fingerprint is a characteristic profile, which is widely applied in the quality control of herbal medicinal products [27]. In 2020, Kharbach et al. [28], reviewed different fingerprint approaches submitted to untargeted and targeted data analysis for the quality and adulteration control of herbal extracts. The application of multivariate data analysis techniques offers a good solution to extract appropriate chemical information from the raw analytical data. The fingerprint approach became a standard method for quality control purposes [29]. The current review discusses the use of fingerprinting techniques on Argan oils, either or not associated with chemometric tools, for untargeted and targeted data analysis applied in the context of quality control, geographical origin determination, and adulteration assessment. 

### 4.1. Spectroscopic Techniques in Quality Control

Nowadays, various spectroscopic techniques, which allow determining complex chemical information on samples from a simple scan, are available. These techniques are based on the interaction of electromagnetic radiation with sample molecules [30]. They have been successfully applied as analysis techniques in several research studies, showing as advantages low operating costs and high speed of analysis [26,27]. In addition, they often are non-destructive, require little or no sample preparation before analysis, and all types of samples (powders, liquids, films, gases, etc.) can be analyzed. These techniques have been applied for both qualitative and quantitative analyses [31].

Spectroscopic techniques apply absorption, emission, or scattering phenomena between electromagnetic radiation and molecules [30]. They have been employed in many fields, such as the agricultural, food, and pharmaceutical [28,29]. The most frequently used are mid-infrared (MIR), near-infrared (NIR), and Raman spectroscopy [32] Figure 2 shows the NIR, MIR and Raman spectra of Argan oil and other vegetable oils, such as corn, rapeseed, sunflower and peanut oil. The data from these techniques need to be submitted to chemometric tools for data trends analysis. Table 2 shows several studies, where spectroscopic techniques have been used to evaluate Argan oil quality.

#### 4.1.1. UV-Visible and Infrared Spectroscopy

Spectroscopic methods associated with chemometric tools for data analysis are frequently used in the quality control and classification of food products. The simplicity of the procedure, speed, accuracy and the low cost of the spectroscopic techniques made UV-Visible and NIR spectroscopy the techniques of choice for the quality control of Argan oil. The information contents of both fatty-acid profiling (by gas chromatography) and UV-Visible fingerprints in combination with chemometric tools (Principal Component Analysis (PCA) and Partial Least Squares Discriminant Analysis (PLS-DA)) were evaluated to classify extra virgin Argan oil (EVAO) samples, produced in five southwestern regions in Morocco, according to their geographical origin, to their extraction process (traditional or mechanical) and to their kernel type (roasted or unroasted) [33]. PLS-DA was performed on the UV-Visible spectra pretreated by the Savitsky Golay smoothing, and first derivative calculation and mean centering. The performances of the constructed models were evaluated by the percentages of sensitivity, specificity and accuracy which achieved 100% for training and prediction sets. UV-Visible spectroscopy is hence an efficient technique to classify Argan oil.

Visible and near infrared spectroscopy (NIR) have also been used as rapid and environmentally friendly techniques to evaluate different quality parameters of Argan seeds (kernels), such as moisture, fatty-acid and seed oil contents and the stability index [34]. PLS models were built using the UV-Visible spectra of Moroccan Argan oil samples to predict the fatty acid contents in oil and seeds. 

FT-MIR spectra with Partial Least Squares Discriminant Analysis were used to discriminate Argan oils from five Moroccan geographical origins (Ait-Baha, Agadir, Essaouira, Tiznit and Taroudant) [20]. Several chemical parameters were also measured, such as free acidity, peroxide value, spectrophotometric indices, fatty-acid composition, tocopherol and sterol contents. The chemical-composition data and the spectroscopic fingerprint, were explored by PCA and then classified by PLS-DA. The five classes were well distinguished. The results based on the chemical composition and on the FT-MIR data were similar. FT-MIR spectroscopy, in combination to PLS-DA, is hence an efficient technique to classify Argan oil from models with a good fit and low prediction errors.

Martín-Ramos et al. [40] analyzed the similarities between the ATR-FT-MIR spectra from *Argania spinosa*, *Rosa rubiginosa* and *Elaeis guineensis* oils. The composition of the three oils was very similar in terms of components, which hampers their identification, but differs in terms of relative proportions, which is reflected in the relative intensities of FT-MIR bands. The main difference was observed from the intensities of the peaks at 966 cm^−1^, which are attributed to the =CH band, and which was higher for the *Rosa rubiginosa* oil than for the other two. As a conclusion from this study, the visual evaluation of the FT-MIR spectra allows obtaining valuable information concerning given bands but shows some limitations concerning the discrimination between the three analyzed oils. However, we believe that better discriminative results would be generated by handling the data with proper statistical and mathematical tools.

#### 4.1.2. Nuclear Magnetic Resonance

Both ¹³C-NMR and ¹H-NMR offer advantages because they are non-invasive and non-destructive techniques used for the determination of organic structures in complex matrices. These techniques have, for instance, been applied to assess the thermal stability of edible oils [41,42], to characterize oils and fats (discrimination from other oils and detection of adulteration) [43], and to estimate quality related properties of Argan oil [32,33,34]. 

Previously, 250 MHz 1H NMR experiments were performed to elucidate the effects of thermal treatment on both roasted- and unroasted-kernel Argan oils, which were kept in the dark at 60 °C for 30 days, in complement to the peroxides and conjugated diene hydroperoxides assessment [35]. Both the aliphatic/diallylmethylene proton ratio (Rad) and the aliphatic/olefinic proton ratio (Rao) were used as indicators to evaluate the resistance of the oil to oxidation. Both ratios remained steady until the 20th day of thermal oxidation. Then after 20 days, Rad increases, whereas Rao remains constant. This change indicates the beginning of the oxidation process. Hamdouch et al. [36] followed the autoxidation of fatty acids of Argan oils stored at 4 °C and analyzed after 6, and 12 months (extracted with hexane) by 1H NMR measurements. The results indicate a decrease in the percentage of unsaturated acids from 76 to 63% and an increase in saturated acids from 24 to 36%. Additionally, 1H NMR and GC analysis to quantify fatty acids in Argan pulp (kernels) were compared. It was found that the main fatty acids are myristic and palmitic acid. The study showed that the 1H NMR could be as efficient (accurate) as the GC technique in term of quantification of fatty acids in Argan pulp.

Recently, 60-MHz benchtop NMR has been used to acquire 1H spectra of Argan oil [37]. A nearest-neighbor method, based on the ensemble of distance metrics in the class analyzed, was developed to evaluate the authenticity of Argan oil. Authentic Argan oil samples were split into training and test set to evaluate the model built as a function of the accept/reject threshold. Furthermore, other vegetable oils were introduced in the model and were all correctly rejected. The results showed the potential of the NMR spectra handled with a nearest-neighbor outlier detection approach for quality and authenticity evaluation of Argan oil.

#### 4.1.3. Inductively Coupled Plasma Atomic Emission Spectroscopy and Optical Emission Spectroscopy

Ennoukh et al. [38] studied the influence of oil extraction methods on the metal contents of the Argan oils. The Argan oils were from four regions, and were extracted with three methods (i.e., traditional, mechanical pressing and solvent extraction). Then, eleven dietary and heavy metals were determined by inductively coupled plasma atomic emission spectroscopy (ICP-AES). The results allowed concluding that the extraction method did not affect the metal content in the Argan oils. 

The content of eight metals (cadmium, chrome, copper, zinc, iron, potassium, magnesium, and calcium) was determined by ICP-AES in roasted (edible) and unroasted (cosmetic) Argan oils collected from four different regions [44]. These elements varied depending on the geographical origin and the production year. When comparing the amounts of these elements in both types of Argan oils, they were found similar, except for calcium, which occurred frequently in lower amounts in unroasted-kernel oils. As a conclusion, the ICP-OES technique demonstrates that the roasting did not influence the elemental content of Argan oil. 

Recently, Mohammed et al. [45] applied ICP-AES to quantify the contents of eleven elements (Ca, P, Mg, Mn, K, Cu, Fe, Cd, Cr, Zn, Sn) in Argan oil, prepared by cold pressing from kernels collected from both fully ripe and unripe fruits. The results show a 55 to 60% metal increase in Argan oil obtained from fully ripe fruit. These findings demonstrate the importance of the maturity level and harvest time in ensuring high nutritional Argan oil quality. Gonzálvez et al. [46] used inductively coupled plasma optical emission spectroscopy (ICP-OES) after microwave-assisted acid digestion to quantify the trace element contents of Argan oil. Moroccan Argan oil showed different concentrations of trace elements (Al, Ca, Cr, Fe, K, Li, Mn, V, and Zn) from other vegetable edible oils. These findings could be beneficial for authentication purposes. The Appendix A summarizes the results for the most important metals quantified in Argan oil samples. In conclusion, the ICP-OES/AES demonstrated its potential to assess the quality of Argan oil based on metal contents, which can vary according to the maturity level, harvest period, geographic origin, or the production year. These findings could be beneficial in identifying the ideal conditions for producing the highest-quality Argan oil.

#### 4.1.4. Electrothermal Atomization–Atomic Absorption Spectroscopy

The extraction method (traditional versus semi-industrial) effect on virgin Argan oil quality was monitored through the metal contents and physicochemical parameters in [47]. Fe, Cu, Cr, Mn and Pb were quantified by electrothermal atomization–atomic absorption spectroscopy (ETA-AAS). The metal levels in the Argan oil samples obtained by the traditional method are higher than those obtained by the semi-industrial method. Those metals, especially Fe, have a deleterious effect on the flavor and oil quality and are involved in oxidation. However, improvements in the technological production process of Argan oil could enhance the oil quality and preserve its chemical content and quality. 

These study findings contradict those of Ennoukh et al. [38], who found with ICP-AES that the extraction procedure has no significant effect on the quality of Argan oil. This inconsistency could be explained by the low quality of the mechanical Argan oil used in that study or by the progress in extraction processes to yield high-quality Argan oil. 

### 4.2. Spectroscopic Techniques in Authentication

Argan oil authentication has become a serious issue which must be carefully considered. Adulteration can result from different fraudulent processes. For instance, it can originate from preparing a mixture by adding cheaper lower-quality oil to authentic oil. Adulteration may also mean selling low quality oil as high quality. Argan oil is susceptible to adulteration, thus the Moroccan agricultural authorities have edited a guideline, describing the chemical parameters and sensorial panel, used as reference for assessing the quality of Argan oil [12]. The Moroccan Normative (N.M. 08.5.090) was published in 2003 to describe the quality specifications of Argan oil and its classification in different categories. Further, the Moroccan authorities have set up an internal protection apparatus by defining the PGI (Protected Geographical Indication) label.

Relevant studies, carried out using spectral fingerprinting techniques and chemometrics, to detect and quantify AO adulteration are listed in Table 3. Some are discussed below.

#### 4.2.1. UV-Visible, Infrared and Raman Spectroscopy

In the presence of HAuCl₄, phenolic acids in Argan oil could be reduced to give AuNPs (gold nanoparticles) as redox products. This reaction leads to color changes measured by UV-Visible spectroscopy at 555 nm. In a study by Zougagh et al. [56], an AuNPs spectrophotometric method was applied to evaluate the adulteration of Argan oil by sunflower and olive oil. This analytical strategy made it possible to classify the samples as positive or negative according to the predefined threshold (Argan oil has a higher amount than the other vegetable oils), thus providing low precision quantification information.

Visible and near infrared spectroscopy (Vis/NIRS) coupled to chemometric tools were applied to detect and quantify Argan oil adulteration with two cheaper vegetable oils purchased from a local market in Oujda, Morocco. The later were first heated to get the same color as that of Argan oil [48]. The spectral data was investigated by PCA and PLS regression. As a first step, with an exploratory PCA model, it was possible to differentiate between pure and adulterated Argan oils samples. Then, a PLS model was built to predict the concentration of adulterant in Argan oil. Good predictive performance parameters were obtained for this model.

The visual differentiation between the FT-MIR spectra of pure and adulterated Argan oil is not possible. To solve this problem, PLS regression was used for modelling and predicting Argan oil adulterated with either sunflower or soybean oil in a range of 0–30%, *w*/*w* [49]. The best PLS model was obtained after applying SNV and smoothing pretreatment. FT-MIR spectra were also handled by chemometric tools by El Orche et al. [50] to quantify Argan oil adulteration with Olive oil. The spectral data were subjected to PLS regression. The PLS model showed acceptable fit and predictive properties. Both studies demonstrated the capability of using PLS-R on FT-MIR spectra to detect adulterants in Argan oil at low concentrations.

Raman spectroscopy, in combination with the hybrid linear analysis method developed by Goicoechea and Olivieri (HLA/GO) [56], was used to quantify Argan oil adulteration with olive oil in the range of 0–20% [51]. The model demonstrated good figures of merit for calibration and prediction with a good sensitivity, specificity, and accuracy. The obtained results demonstrated that the combination of Raman spectroscopy with the HLA/GO- model represents an analytical tool for determining the concentration of olive oil in Argan oil.

Raman spectroscopy, as technique, is widely used for quality control and safety determinations of other edible oil products, but a little less for Argan oil. Thus, Raman spectroscopy could be more exploited in this field since intrinsically it is an interesting technique [32,57,58,59].

#### 4.2.2. Nuclear Magnetic Resonance

¹H NMR was employed for the determination of Argan oil adulterated with other vegetable oils (sunflower, avocado, rapeseed, sweet almond) [37]. This technique provides useful information about monounsaturated, polyunsaturated, and saturated fatty acids, and about some minor compounds (polyphenols, tocopherols, sterols…) of Argan oils. A nearest-neighbor class model (KNN), exploiting the whole spectral range, was able to discriminate between pure and adulterated Argan oils, with a detection limit around 20%. This model was challenged with various vegetable oils and adulterated samples were all correctly rejected. In conclusion, in addition to the chemical information extracted by this technique, its combination with KNN can go further and detect the adulteration of Argan oil.

#### 4.2.3. Fluorescence Spectroscopy

Fluorescence spectroscopy data were handled with chemometric tools for the analysis of Argan oil adulterated with olive oil [37]. Both PCA and PLS were applied on the spectral data. The PLS model for the prediction of the Argan oil adulteration content with olive oil showed a coefficient of determination of R² = 0.992 (predicted vs. real content). This approach exhibited its capacity for the detection of adulteration with a sensitivity starting from 0.43% of olive oil mixed to Argan oil (*w*/*w*). 

Synchronous fluorescence spectroscopy (SFS) was used to detect and quantify Argan oil, adulterated by corn oil from 0.5 to 10.0% (*m*/*m*) [53]. This technique requires the determination of a useful wavelength interval, which was determined in this study at ten Δλ intervals. Two calibration approaches were assessed on the fusion of the SFS data obtained at the 10 Δλ intervals. One is PLS multivariate calibration, the second is a univariate calibration, known as the area under the curve of the spectral bands (AUC). Globally, the results showed that the PLS regression model and the AUC approach both provide equivalent prediction errors.

#### 4.2.4. Inductively Coupled Plasma Optical Emission Spectroscopy

Inductively coupled plasma optical emission spectrometry (ICP-OES) data was subjected to multivariate data analysis to detect adulteration of Argan oil by cheaper vegetable oils (sunflower, olive, seeds and soya oils) [54]. The contents of sixteen elements were used as variables for data analysis, i.e., hierarchical cluster analysis (HCA), principal component analysis (PCA), classification trees by chi-squared automatic interaction detector (CHAID), and discriminant analysis (DA). As a result, HCA allowed discriminating sunflower oil from the others. On the contrary, PCA separated the data in three distinct groups (sunflower, Argan, and a third group which gathers olive, seeds and soya oils), whereas CHAID and DA distinguished between all studied oils. DA was indicated as a successful tool to detect Argan oil adulteration.

### 4.3. Separation Techniques in Quality Control

Chromatographic techniques provide powerful analytical methods for the separation and quantitative determination of macro- and micro molecules with highly similar chemical structures [60]. These techniques are coupled to appropriate detectors and nowadays, chromatographic data, subjected to data-handling and chemometric approaches, are widely applied for quality control [61]. Several chromatographic techniques were investigated for the quality control of Argan oil and were applied for targeted analysis. Interesting studies are listed in Table 4 and shortly discussed in the following sections.

#### 4.3.1. Gas Chromatography

Argan oil purity should be examined to verify its authenticity. Rezanková et al. [63] established a gas chromatographic method to target the fatty-acid and triacylglycerol composition in Argan oils (targeted analysis) and to compare Argan oils from the shop and the pharmacy, with other vegetable oils prepared in the laboratory (olive, palm, sunflower, and soy oils). Cluster analysis, PCA, and chi-square dissimilarity were applied on the results to differentiate between the Argan and other vegetable oils. 

Recently, GC fatty-acid profiling was used to discriminate Argan oils from different Moroccan regions (Essaouira, Taroudant, Tiznit, Aït-Baha and Sidi-Ifni) [1]. The fatty-acid composition was subjected to discriminant analysis (DA) and orthogonal projections to latent structures- discriminant analysis (OPLS-DA). The OPLS-DA results exhibited a correct classification rate of 92%. Fatty-acid trans-isomer limits for authenticity and quality control of Argan oil could be implemented. Kharbach et al. [33] also highlighted the feasibility of fatty-acid profiling (issued by GC-FID) in combination with the PLS-DA classification method for geographical-origin, extraction-method and kernel-type discrimination. 

In conclusion, formulated from those studies, fatty-acids profile quantification in combination with multivariate methods may provide an effective way to verify the authenticity and provenance of Argan oil, and to distinguish Argan oil from other vegetable oils. Regarding the reviewed studies, the GC technique was only applied as targeted approach and the quantified chemical profiles are used in the multivariate data analysis.

#### 4.3.2. High-Performance Liquid Chromatography

HPLC was used by Matthäus et al. [54] to evaluate the effects of the roasting-process and storage conditions on the stability and quality of Argan oil. Tocopherol and sterol concentrations were determined, using HPLC-fluorescence detection and GLC-FID, in both traditionally pressed roasted kernels and mechanically pressed unroasted and roasted kernels. The levels of tocopherol and sterol are found higher in Argan oils prepared from mechanically pressed roasted kernels than from the traditional method, which confers a better stability to Argan oil extracted by the mechanical process. 

In another study, Hilali et al. [55] determined the fatty-acids, tocopherol, and triglyceride composition as well as quality parameters in Argan oils obtained from different extraction methods, mechanical pressing from roasted or unroasted kernels, and from different origins (Tidzi, Beniznassen, Ait mzal, and Ighrem). They used GC-FID on a capillary column, HPLC on a silica column with a fluorimetric detection and on a reversed phase C18 column with refractometric detection for fatty-acids, tocopherols and triglycerides analysis, respectively. The results showed that the extraction method and the origin may affect the peroxide index, the rate of unsaponifiable matter as well as the fatty-acid, tocopherol and triglyceride compositions.

In Gharby et al. [56], the evolution of the quality of Argan oil was followed in order to evaluate the impact of storage conditions and of roasting the kernels. Tocopherol composition was quantified by HPLC coupled to fluorescence detection, and the fatty-acids were analyzed by gas chromatography equipped with an FID. This study confirmed what is quoted above: different storage conditions had a signification effect on the Argan oil quality. For industry, it is useful to know which storage conditions are better to maintain a superior quality and optimal conservation of Argan oil.

HPLC was also used to evaluate the impact of filtering cold pressed Argan oil from unroasted kernels on the quality and stability [71]. Filtration was found to have a significant negative effect on the oxidative stability, phospholipid and peroxide values. However, it did not affect the tocopherol levels or the fatty-acid and sterol composition.

In Atifi et al. [67], another parameter affecting Argan oil quality, i.e., fruit maturity, was studied. Oil extracted at three levels of fruit maturity (over ripe, ripe, and unripe) was analyzed to evaluate the effect of maturity on the quality, quantity and chemical composition of Argan oil. Sterol content was found increased with maturity level, fatty-acids content increased in ripe and decreased in over ripe fruit. This knowledge may help determining the proper time of harvesting in relation to the Argan oil composition wanted and leads to a better valorization of the Argan tree.

In recent years, the analytical data gathered to evaluate Argan oil quality are increasingly combined to chemometric tools for qualitative and quantitative purposes. In Rueda et al. [73], HPLC with diode-array and fluorescence detection was used to characterize and quantify the phenolic profile and the tocopherols in Argan oil and eight other vegetable oils. As chemometric method, multivariate factor analysis (MFA) was applied to evaluate similarities and differences between extra virgin Argan oil and the other edible vegetable virgin oils. The relationships between the phenolic compounds and tocopherols for each oil were also assessed by evaluating the correlation coefficients. The MFA revealed that γ-tocopherol shows the strongest influence in the discrimination between the different oils. 

As a summary of this section, it can be concluded that chromatographic methods are the best for identifying the various components of Argan oil. However, combining the chromatographic data with chemometric tools may offer more convenience to predict the purity of Argan oil. Chromatography is an effective technique for the quality control of Argan oil based on the identification and measurement of small components. In terms of accuracy, selectivity, and sensitivity, it is regarded as a suitable technique. Unfortunately, it cannot be considered as environmentally friendly because it requires the use of organic solvents for sample preparation and analysis. Therefore, it is necessary to review this procedure, come up with alternative sample preparation techniques, and change to more environmentally-friendly techniques, such as supercritical fluid chromatography (SFC), which has not yet been used for the analysis and quality control of argan oil.

### 4.4. Separation Techniques in Authentication

Targeted analyses are reported for the detection of Argan oil adulteration. They are based on qualifying or quantifying specific compounds (markers) or groups of molecules, such as triglycerides, tocopherols, fatty acids, sterols, or polyphenols. These markers are determined by gas (GC) and/or liquid chromatography (LC) coupled to different detectors. Representative chromatograms of triacylglycerols from Argan, sunflower, olive, and soybean oils are shown in Figure 3. Relevant studies carried out using targeted chromatographic techniques and chemometric methods to detect Argan oil adulteration are listed in Table 5. Some are discussed in the section below.

No application of chromatographic data in untargeted analysis was found.

Campesterol is a sterol present in Argan oil in a percentage below 0.4% of the total sterols [81]. However, campesterol occurs in high percentages in soybean oil (16–24% of total sterols), rapeseed oil (25–39%), and sunflower oil (7.5–13%). In the study by Hilali et al. [81], the campesterol level was quantified by GC-FID and then used as an adulteration marker for Argan oil. The study establishes 0.4% campesterol as maximal level, and adulteration of Argan oil with vegetable oils could be determined with campesterol levels between 1 and 5%. 

In a study by Ourrach et al. [82], some markers were selected to determine Argan oil adulteration by sunflower and virgin olive oils. The results, obtained by gas chromatography, showed that 3,5-stigmastadien, kaurene and pheophytin-a can be used as markers to detect adulteration up to 5%. Because of the similarities in fatty acids between Argan oil and the adulterants, the GC-FID technique could detect adulteration only above 10%. Furthermore, stigmastadiene has been shown to be a potential marker for detecting adulteration up to 3%.

In Salghi et al. [80], the triacylglycerol profile was established by HPLC with evaporative light scattering detection (ELSD) and proposed for detecting Argan oil adulteration up to a level of 5% by sunflower, soybean, and olive oils. The developed method, quantifying triacylglycerols as markers, showed a high selectivity and sensitivity. In a study by Pagliuca et al. [83], the triacylglycerol profile was determined by both UHPLC and HPTLC to detect counterfeit Argan oil and Argan-oil-based products, like creams and oils. A similar result to that of the prior study was obtained using the UHPLC method, enabling the detection of up to 5% of the adulterant in Argan oil. However, the HPTLC method, used to determine the amount of Argan oil in cosmetic samples, was able to quantify up to 0.5% of Argan oil in Argan-oil-based products.

### 4.5. Other Techniques

Both roasted and unroasted-kernel Argan oils, adulterated with sunflower oil, were studied using a combination of an electronic nose (e-nose) and a voltammetric electronic tongue (e-tongue) [84]. Three multivariate data analysis techniques, PCA, Discriminant Factor Analysis (DFA), and SVM, were applied on the acquired data. PCA and DFA applied on either e-nose or e-tongue data distinguished pure Argan oil, sunflower oil and Argan oil adulterated with sunflower oil (10 to 70% of sunflower oil). On the other hand, SVM modelling on e-nose data, applied to develop a classification model according to the percentage of adulteration, achieved acceptable success rates; 92 and 83% for the recognition of roasted and unroasted Argan oil adulterated with 10 to 70% sunflower oil, respectively, while SVM applied on e-tongue attained 100% success rate in the recognition of both roasted and unroasted Argan-oil adulteration. The study confirmed that the use of e-nose and e-tongue methods with multivariate data analysis techniques showed an acceptable effectiveness to detect the adulterants in Argan oil based on metal oxide semiconductor sensors.

## 5. Conclusions

This review provides an overview of research studies in relation to the quality control and authentication of Moroccan Argan oil, mainly using spectroscopic and separation techniques. Chemometric tools were occasionally used to extract useful information from the data acquired by the analytical techniques.

Targeted chromatographic approaches are based on specific markers, such as triacylglycerols, fatty acids, sterols, tocopherols and polyphenols. They have been used in many research studies to ensure the quality and authenticity of Argan oil. 

Untargeted approaches, ranging from typical spectroscopic (NMR, NIR, MIR, UV-Visible, fluorescence and Raman spectroscopic) to less commonly used techniques (electronic nose or sensors), without excluding chromatographic techniques, have also demonstrated their suitability for evaluating Argan oil quality and detecting adulteration, through combining these techniques to chemometric algorithms in order to handle the complex data matrices. Despite the benefits of these combinations and their wide range of applications, the Moroccan Normalization Guidelines and EU regulations do not include chemometrics in their guidelines, though chemometrics may provide additional tools to assess the quality and authenticity of Argan oil. Therefore, targeted and untargeted approaches combined with chemometrics will increasingly be used, and it is envisaged that they will be included in official regulations in the coming years.

## Figures and Tables

**Figure 1 molecules-28-01818-f001:**
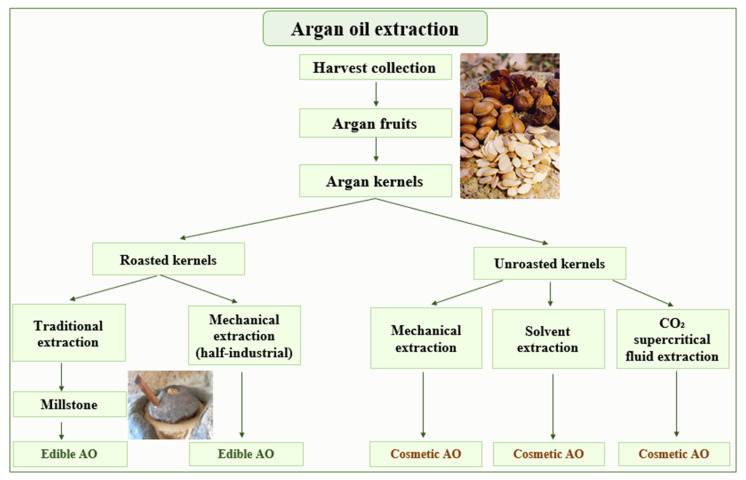
Methods used for Argan oil extraction.

**Figure 2 molecules-28-01818-f002:**
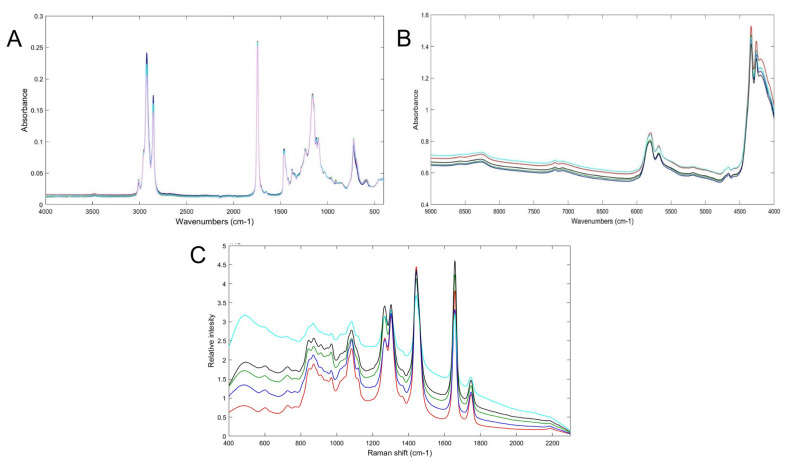
(**A**) FT-MIR, (**B**) FT-NIR and (**C**) Raman spectra of pure Argan oil (red/pink), corn oil (green), rapeseed oil (sky blue), sunflower oil (black) and peanut oil (blue). Spectra measured by authors.

**Figure 3 molecules-28-01818-f003:**
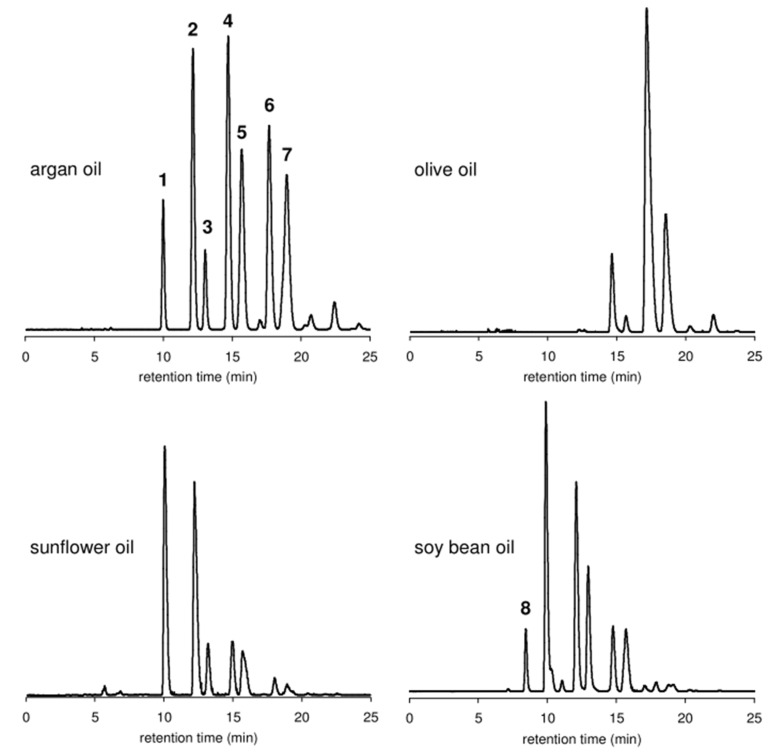
HPLC-ELSD chromatograms of triacylglycerols from Argan oil, olive oil, sunflower oil, and soybean oil (peaks were not identified). Reproduced with permission from Ref. [80].

**Table 1 molecules-28-01818-t001:** Percentages of total fatty acids in Argan oil and other vegetable oils.

	Vegetable Oil						
Fatty Acid	Argan [17,18]	Olive [21]	Soybean [22]	Corn [23]	Sunflower [24]	Peanut [24]	Rice [24]
Oleic C18:1	43.0–49.0	66.4–78.6	23.2–23.9	8.5–46.1	15.0–40.0	23.0–41.0	30.0–45.0
Linoleic C18:2	29.3–36.0	5.5–11.8	53.30–55.9	36.6–66.8	40.0–74.0	15.0–48.0	35.0–50.0
Stearic C18:0	4.2–7.2	1.9–3.0	4.0–4.4	0.9–4.5	2.5–7.0	2.0–5.0	1.0–2.5
Palmitic C16:0	11.5–15.0	11.6–16.5	9.6–10.3	6.3–18.2	5.0–8.0	8.0–13.5	17.0–22.0

**Table 2 molecules-28-01818-t002:** Spectroscopic techniques used for the quality control of Argan oil.

Study Purpose	Method	Reference
Geographical origin characterization and classification of Argan oils	FT-MIR	[20]
Shelf life of extra virgin Argan oil	FT-MIR	[25]
Geographical origin classification of EVAO according to the extraction process/kernel type	UV-Vis	[33]
Evaluating Argan seed quality	Vis/NIR	[34]
Oxidative stability of Argan oil	NMR	[35]
Evaluation of the Argan oil fatty acids auto-oxidation	NMR	[36]
Screening the quality and authenticity of AO	¹H NMR	[37]
Extraction method effects on Argan oil quality	ICP-AES	[38]
Geographic traceability of Moroccan Argan oils	SIFT-MS	[39]

**Table 3 molecules-28-01818-t003:** Spectroscopic techniques and chemometric tools applied for adulteration evaluation of Argan oil: untargeted analysis.

Aim	Fingerprinting Technique	Chemometric Tools	References
Adulteration of Argan oil with sunflower, avocado, sesame, rapeseed and sweet almond oil	¹H-NMR	Nearest-neighbor outlier detection	[37]
Argan oil adulteration with cheaper vegetable oils	Visible/NIR	PCA, PLS	[48]
Analysis of Argan oil adulteration with sunflower or soybean oil	FT-MIR	PLS	[49]
Detection and quantification of Argan oil adulteration with olive oil	FT-MIR	PLS	[50]
Detection of olive oil in Argan oil	Raman	HLA	[51]
Detection of Argan oil adulteration with olive oil	Fluorescence spectroscopy	PCA, PLS	[52]
Argan oil adulteration with corn oil	Fluorescence Spectroscopy	PLS	[53]
Oils classification based on their elemental contents for the detection of adulteration of Argan oil with sunflower, olive, seeds and soya oil	ICP-OES	PCA, HCA, DA	[54]
Geographical classification of Moroccan Argan oils and the rapid detection of its adulteration	ATR-FT-MIR	PCA, SIMCA, DD-SIMCA, PLS	[55]

**Table 4 molecules-28-01818-t004:** Chromatographic techniques for the quality control of Argan oil: targeted analysis. ANOVA: Analysis of Variance; GLM: General Linear Model, NA: not applicable.

Aims	Techniques	Data Analysis Methods	References
Effect evaluation of clones and age, year of harvest and geographical origin on Argan oil quality	GC-FID NP-HPLC-FLD	ANOVA PCA	[2]
Comparison between two methods of Argan oil extraction	RP-HPLC-DAD-MS	Mean values ± standard deviation	[62]
Fatty-acid profile to evaluate the authenticity of cosmetic Argan oil	GC-FID	ANOVA, PCA, DA, OPLS-DA	[1]
Identification of fatty acids and triacylglycerols	GC-FID	Chi-square test	[63]
Effect of processing on edible Argan oil quality	RP-HPLC-FD, GLC-FID	*t*-test	[64]
Evaluation of extraction method, origin of production and altitude on Argan oil composition	RP-HPLC-FD, GC-FID	NA	[65]
Effect of storage conditions and kernel roasting on Extra Virgin Argan Oil quality	RP-HPLC-FD GC-FID	ANOVA	[66]
Effect of Argan fruit maturity (over ripe, ripe, and unripe) on the quality, quantity and chemical composition of Argan oil	GC-FID	ANOVA	[67]
Effect of Argan kernel storage conditions on Argan oil quality	GC-FID	*t*-test	[68]
Effect of harvest date on Argan oil quality	RP-HPLC-FD GC-FID	ANOVA	[69]
Oxidative stability of edible Argan oil	RP-HPLC-FD	*t*-test	[70]
Effect of filtration on virgin Argan oil quality and stability	NP-HPLC-FD	ANOVA	[71]
Influence of Argan fruit peel on the quality and stability of Argan oil	RP-HPLC-FD GC-FID	NA	[72]
The characterization of Extra Virgin Argan oil	RP-HPLC-FD	MFA	[73]
Effect of roasting temperature and time on Argan oil stability	RP-HPLC-FD	ANOVA	[74]
Evaluation of authenticity and quality of Argan oils	RP-HPLC-ELSD	PCA, clustering	[75]
Evaluation of the fatty-acid composition and oil contents in trees with different morphological characters and geographical localization	GC-FID	GLM, ANOVA	[76]
Optimization of roasting conditions of Argan kernels for high quality edible Argan oil	RP-HPLC- FD	ANOVA, CCD	[77]
Determination of geographical origin of Argan oil using fatty-acid profiling	GC-FID	PCA, PLS-DA	[33]
Comparison of triacylglycerol profiles of oils	UPLC-ESI-MS	ANOVA	[78]
Determination of antioxidant activity, total phenolics and fatty acids	GC-FID	ANOVA	[79]
Quality of cosmetic Argan oil extracted by supercritical fluid extraction	GC-FID, HPLC UV-Vis	NA	[14]

**Table 5 molecules-28-01818-t005:** Chromatographic techniques to detect adulteration in Argan oil: targeted analysis.

Adulterants	Markers	Separation Techniques	Data Analysis Methods	References
Sunflower and olive oil	Total phenolic acids	HPLC-DAD and fluorescence detector	*t*-test	[56]
Soybean, rapeseed, sunflower, apricot, arachis and hazelnut oil	Campesterol	GC-FID	ANOVA, Tukey test	[81]
Sunflower, olive, and soybean oil	Triacylglycerols	HPLC-ELSD	*t*-test	[80]
Sunflower and virgin olive oil	Fatty acids 3,5-stigmastadiene, kaurene and pheophytin-a	GC-FID HPLC-DAD	ANOVA, Dunnett’s test	[82]
Almond, coconut, linseed, wheat germ, sunflower, peanut, olive, soybean, rapeseed, hemp oils and shea butter	Triacylglycerols	UHPLC-PDA-ESI-TOF/MS HPTLC	ANOVA, Dunnett’s test	[83]

## Data Availability

Not applicable.

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
