# Peer review of "Quality Control and Authentication of Argan Oils: Application of Advanced Analytical Techniques"

_molecules, 2023, doi:10.3390/molecules28041818_

Round 1

Reviewer 1 Report

This review provides a well overview of research studies in relation to the quality control and authentication of Moroccan Argan oil, mainly using spectroscopic and separation techniques. Chemometric tools were also introduced to extract useful information from the data acquired by the analytical techniques.  This review is well organized but if some suggestions are accepted to revise this manuscript is better.

1  Many spectroscopic methods are proposed, but if authors listed the typical figures of spectra can help readers get known well.

2  Some commercial application methods to control the quality of Argan oil should be introduced.

Author Response

1- Many spectroscopic methods are proposed, but if authors listed the typical figures of spectra can help readers get known well.

Response: As suggested by the reviewer, some figures were added.

Fig 2. (A) FT-MIR, (B) FT-NIR and (C) Raman spectra of pure Argan oil, sunflower oil, rapeseed oil, corn oil and peanut oil. (See line: 204)

Fig 3. HPLC-ELSD chromatograms of Argan oil, olive oil, sunflower oil, and soybean oil) [ reproduced with permission from Ref. 31]. (See line: 534)

2- Some commercial application methods to control the quality of Argan oil should be introduced.

Response: The quality control of Argan oil is based on its chemical composition described in the restrictions imposed by a Moroccan normative directive (N.M. 08.5.090). We did not find commercial application approaches (similar to those described in this standard).

Reviewer 2 Report

The authors adequately review analytical techniques and chemometric tools used for the authentication and quality control of Argan oil, an important natural product. The main comment consists of the need for higher quality figures. This extends to the photos shown in Fig. 1, and the spectrum and chromatogram shown in Fig. 2. Most importantly, more figures (ideally, one for each technique instead of a single Fig. 2) would be of particular interest to complement/illustrate the explanations in the text. Although this requires re-printing published figures (and thereby asking for such rights), the benefits would warrant this.

31: The family to which Argania belongs (Sapotaceae) may be of interest to biologists.

Fig. S1: In the caption, cite the source of the plotted distribution data for the Figure to be understood on its own.

88-91,121-122: How is the chemical composition in the oil obtained by SFE different from other extraction methods?

134: "healt"→"health"

144-155: The position of the unsaturated bond has a major impact on the dietary impact of unsaturated fatty acids. Adding the position (ω-6) will be of interest to many readers.

142 ff and S1: Does the fatty acid composition described in this paragraph (and Tables 1 and S1) refer to the total fatty acids (esterified and free) or only to the free fatty acids? Are free fatty acids a significant component within the 1% outside the TAG fraction (99%)?

Table 1: Since it is already contained in the first line ("Vegetable oil"), removing the "oil" from the second line ("Argan oil," "Olive oil," etc.) may get rid of the line break in the column "Sunflower oil."

157: What is the dietary evaluation of the saturated stearic acid, and how does this compare to the proposed health benefits of Argan oil?

Fig. 2: This figure requires further elaboration. E.g., what do the numbers in Fig. 2B refer to?
Instead of one figure, individual figures appearing in the respective sections of the manuscript would be beneficial. As placed now, Fig. 2 may appear as a (low resolution) stand-in for FT-MIR and HPLC analysis in general rather than pertaining to a review of Argon oil.

224ff, 233ff: [18] and [35] use MIR, but the title and several passages of this section speak exclusively of NIR.

248: "to assess the thermal stability..., to characterize oils and fats..., and to estimate quality related properties of Argan oil....". Which aspects (outside thermal stability and quality related properties) were characterized?

263"...NMR could be as efficient as the GC technique in term of quantification of fatty acids..." In chromatography, separation efficiency has a specific meaning that does not translate to spectroscopy. What exactly is meant by "efficient" in this context? 

Table S2: Remove unwanted line breaks and empty lines

Table 3:
-in the line of [52], it should be specified which wave length of IR was used in that study (i.e., MIR)
-in the line of [30], the authors specify "FT-MIR." In the rest of the table, the use of FT is not specified (even when it was used such as in [52])

335-341: If the phenolic acid content is ultimately responsible for the differentiation between pure Argan oil and vegetable oil adulterated Argan oil: Are there more or less phenolic acids in Argan oil?

Table 4:
-The phase of the HPLC method [1] is further characterized as normal phase (NP-HPLC). Why is this only done for [1] but not for the other instances of HPLC? If this implies RP-HPLC in all other cases, please make this explicit.

-Similarly, the column of the GC method [66] is further characterized as capillary column. But not all other GC methods used packed columns.

-Some redundancies can be eliminated (e.g.,"Determination of...with PLS-DA" in the line of [38], since PLS-DA are already stated in the column for data analysis methods) and abbreviations (ELSD for evaporative light-scattering detector – especially since this abbreviation is used later in the manuscript; FD for fluorescence detection) may free up space.

-It is difficult to delineate the different studies when they span more than one line. Top alignment or add delimiting lines may help.

435-436: "...and the quantified chemicals are used in the data analysis." Does this mean quantitative data was required, while qualitative data by itself was insufficient? Please clarify.

487-491: The authors describe the need for greener chromatographic methods for Argan oil analysis. Previously, SFE was described as a favorable extraction method. This raises an interesting question: are there no reports of SFC methods for the characterization of Argan oil?

General:

-Figures: The resolution is too low. The Figures in the separate Figure file have a higher resolution than the ones imported into the manuscript. The size of the photos in Fig. 1, esp. of the millstone, is too low. The other photos should be explained in more detail in the caption.
-Superfluous space between words (e.g., 167, 249)
-Superfluous commas in "author et al., [citation]" (e.g., 294)
-Italicize physical abbrevations (m, w)
-Use non-breaking space to prevent line breaks between number and unit (e.g., line break in 291-292, no space in 251).

Author Response

1- 31: The family to which Argania belongs (Sapotaceae) may be of interest to biologists.

The sentence was rephrased according to the reviewer’s recommendation. “The Argan tree (Argania spinosa. L), belonging to the Sapotaceae family, is a topical plant appearing endemically in southwestern Morocco.” (See lines: 31-32)

2- S1: In the caption, cite the source of the plotted distribution data for the Figure to be understood on its own.

We developed the plot of Fig. S1 ourselves, based on what we discovered in the literature.

3- 88-91,121-122: How is the chemical composition in the oil obtained by SFE different from other extraction methods?

“The physicochemical parameters of the extracted oils obtained by supercritical fluid extraction from (SFE) and traditional methods are comparable. The SFE, as a technique used for oil processing, does not therefore markedly alter the quality of Argan.” (See lines: 92-94)

4- 134: "healt"→"health"

Error was corrected. (See line: 137)

5- 144-155: The position of the unsaturated bond has a major impact on the dietary impact of unsaturated fatty acids. Adding the position (ω-6) will be of interest to many readers.

We agree with the reviewer. The sentence has been rewritten to add the position of the unsaturated bond. “They are essential for health promotion through the diet [1]. The predominant unsaturated fatty acids are oleic ω-9 (43-49 %) and linoleic acid ω-9 (29-36 %).” (See lines: 147-148)

6- 142 ff and S1: Does the fatty acid composition described in this paragraph (and Tables 1 and S1) refer to the total fatty acids (esterified and free) or only to the free fatty acids? Are free fatty acids a significant component within the 1% outside the TAG fraction (99%)?

The fatty acid composition described in this paragraph (and Tables 1 and S1) refer to the total fatty acids.

7- Table 1: Since it is already contained in the first line ("Vegetable oil"), removing the "oil" from the second line ("Argan oil," "Olive oil," etc.) may get rid of the line break in the column "Sunflower "

Thank you for your comment. This was corrected accordingly.

8- 157: What is the dietary evaluation of the saturatedstearic acid, and how does this compare to the proposed health benefits of Argan oil?

Discussing the dietary evaluation of the saturated stearic acid (or of any other fatty acid), and how this compares to the proposed health benefits of Argan oil is outside the scope of this paper. Therefore, such discussion was not added to the paper.
Stearic acid only was mentioned in the paper because its difference in amount allowed distinguishing it from some other vegetable oils.

9- 2: This figure requires further elaboration. E.g., what do the numbers in Fig. 2B refer to? Instead of one figure, individual figures appearing in the respective sections of the manuscript would be beneficial. As placed now, Fig. 2 may appear as a (low resolution) stand-in for FT-MIR and HPLC analysis in general rather than pertaining to a review of Argon oil.

As suggested by the reviewer, to increase the resolution of this figure, we divided it into two figures. A first containing our own FT-NIR, FT-MIR and Raman spectra (Fig 2), and a second containing the chromatograms of Argan oil and other vegetable oils (Fig 3).

10- 224ff, 233ff: [18] and [35] use MIR, but the title and several passages of this section speak exclusively of N

We agree with the reviewer,. We removed (NIR) from the title. ‘’4.1.1. UV-Visible and Infrared spectroscopy’’ (See line: 229)

11- 248"to assess the thermal stability..., to characterize oils and fats..., and to estimate quality related properties of Argan oil....". Which aspects (outside thermal stability and quality related properties) were characterized?

In Fang et al. (2013), (Characterization of oils and fats by 1H NMR and GC/MS fingerprinting: Classification, prediction and detection of adulteration), NMR and GC/MS fingerprints have been used in combination with chemometric tools to characterize different types of oils and fats such as sunflower, corn, coconut and chicken fats... The oils were distinguished and adulteration was detected. (See line: 274-275)
This was added to the text.

12- 263"...NMR could be as efficient as the GC technique in term of quantification of fatty acids..." In chromatography, separation efficiency has a specific meaning that does not translate to spectroscopy. What exactly is meant by "efficient" in this context? 

In this context, “efficient” means accurate, given that the results from both methods were comparable. This was clarified in the text.

13- Table S2: Remove unwanted line breaks and empty lines.

Unwanted line breaks were removed. (See Table S2)

14- Table 3:
-in the line of [52], it should be specified which wave length of IR was used in that study (i.e., MIR)
-in the line of [30], the authors specify "FT-MIR." In the rest of the table, the use of FT is not specified (even when it was used such as in [52])

Suggested changes were made. (See Table 3)

15- 335-341: If the phenolic acid content is ultimately responsible for the differentiation between pure Argan oil and vegetable oil adulterated Argan oil: Are there more or less phenolic acids in Argan oil?

Argan oil has a higher percentage of phenolic acids than the other vegetable oils used in this study. This was clarified in the text. “Argan oil has a higher amount than the other vegetable oils” (See lines 365-366)

16- Table 4:
-The phase of the HPLC method [1] is further characterized as normal phase (NP-HPLC). Why is this only done for [1] but not for the other instances of HPLC? If this implies RP-HPLC in all other cases, please make this explicit.

As suggested by the reviewer, the chromatographic mode has been added to table 4. (See Table 4)

-Similarly, the column of the GC method [66] is further characterized as capillary column. But not all other GC methods used packed columns.

Thank you for this remark. In all studies, a capillary column was used.

Some redundancies can be eliminated (e.g.,"Determination of...with PLS-DA" in the line of [38], since PLS-DA are already stated in the column for data analysis methods) and abbreviations (ELSDfor evaporative light-scattering detector – especially since this abbreviation is used later in the manuscript; FDfor fluorescence detection) may free up space.

As suggested by the reviewer, redundancies have been eliminated. (See Table 4)

It is difficult to delineate the different studies when they span more than one line. Top alignment or add delimiting lines may help.

We agree with the reviewer. To distinguish between the various studies, table 4 has undergone minor adjustments. (See Table 4)

17- 435-436: "...and the quantified chemicals are used in the data analysis." Does this mean quantitative data was required, while qualitative data by itself was insufficient? Please clarify.

Quantitative data are essential since they provide quantitative information on the various oils' component. However, a more complex matrix is produced when we quantify many substances simultaneously while using multiple samples (e.g., different type of oils). In this case, chemometric tools such as Cluster analysis, PCA, which Rezanková et al., 1999 utilized in their paper, or DA and OPLS-DA as in Miklavi et al., 2020, make it easier to analyze and extract relevant information from the data. This is what we call targeted analysis.

18- 487-491: The authors describe the need for greener chromatographic methods for Argan oil analysis. Previously, SFE was described as a favorable extraction method. This raises an interesting question: are there no reports of SFC methods for the characterization of Argan oil?

Greener and environmentally friendly chromatographic techniques are required for the analyses of Argan oil. However, to the best of our knowledge, no study has characterized and evaluated the quality of Argan oil using supercritical fluid chromatography.

Round 2

Reviewer 2 Report

The authors have addressed most concerns raised in the original comments, improving the quality of the manuscript considerably. Addressed comments are shown in green.

However, major issues remain or have been brought up by the introduction of the new figures. Remaining comments are shown in red.

1- 31: The family to which Argania belongs (Sapotaceae) may be of interest to biologists.

The sentence was rephrased according to the reviewer’s recommendation. “The Argan tree (Argania spinosa. L), belonging to the Sapotaceae family, is a topical plant appearing endemically in southwestern Morocco.” (See lines: 31-32)

The issue has been resolved.

2- S1: In the caption, cite the source of the plotted distribution data for the Figure to be understood on its own.

We developed the plot of Fig. S1 ourselves, based on what we discovered in the literature.

This is the point: without citing the literature, the veracity of the plotted data cannot be confirmed.

Please cite the discovered literature.

3- 88-91,121-122: How is the chemical composition in the oil obtained by SFE different from other extraction methods?

“The physicochemical parameters of the extracted oils obtained by supercritical fluid extraction from (SFE) and traditional methods are comparable. The SFE, as a technique used for oil processing, does not therefore markedly alter the quality of Argan.” (See lines: 92-94)

The newly added paragraph contradicts the previous paragraph:
"...
to obtain the best chemical composition 90 of the extra virgin oil, extraction by COâ‚‚ under supercritical conditions is also performed..." contradicts "...the physicochemical parameters ... are comparable. The SFE [note: it is uncommon to put an article in front of "SFE"] ... does not ... alter the quality...". 

Please clarify how the chemical composition is better while the quality is the same.

4- 134: "healt"→"health"

Error was corrected. (See line: 137)

The issue has been resolved.

5- 144-155: The position of the unsaturated bond has a major impact on the dietary impact of unsaturated fatty acids. Adding the position (ω-6) will be of interest to many readers.

We agree with the reviewer. The sentence has been rewritten to add the position of the unsaturated bond. “They are essential for health promotion through the diet [1]. The predominant unsaturated fatty acids are oleic ω-9 [ω-6] (43-49 %) and linoleic acid ω-9 (29-36 %).” (See lines: 147-148)

The issue has been resolved.

6- 142 ff and S1: Does the fatty acid composition described in this paragraph (and Tables 1 and S1) refer to the total fatty acids (esterified and free) or only to the free fatty acids? Are free fatty acids a significant component within the 1% outside the TAG fraction (99%)?

The fatty acid composition described in this paragraph (and Tables 1 and S1) refer to the total fatty acids.

The issue has been resolved.

7- Table 1: Since it is already contained in the first line ("Vegetable oil"), removing the "oil" from the second line ("Argan oil," "Olive oil," etc.) may get rid of the line break in the column "Sunflower "

Thank you for your comment. This was corrected accordingly.

The issue has been resolved.

8- 157: What is the dietary evaluation of the saturated stearic acid, and how does this compare to the proposed health benefits of Argan oil?

Discussing the dietary evaluation of the saturated stearic acid (or of any other fatty acid), and how this compares to the proposed health benefits of Argan oil is outside the scope of this paper. Therefore, such discussion was not added to the paper.
Stearic acid only was mentioned in the paper because its difference in amount allowed distinguishing it from some other vegetable oils.

The issue has been resolved.

9- 2: This figure requires further elaboration. E.g., what do the numbers in Fig. 2B refer to? Instead of one figure, individual figures appearing in the respective sections of the manuscript would be beneficial. As placed now, Fig. 2 may appear as a (low resolution) stand-in for FT-MIR and HPLC analysis in general rather than pertaining to a review of Argon oil.

As suggested by the reviewer, to increase the resolution of this figure, we divided it into two figures. A first containing our own FT-NIR, FT-MIR and Raman spectra (Fig 2), and a second containing the chromatograms of Argan oil and other vegetable oils (Fig 3).

This change has greatly improved the quality of the manuscript. However, the newly created Figure 2 requires further elaboration: as provided in the caption, the figure shows "(A) FT-MIR, (B) FT-NIR and (C) Raman spectra of pure Argan oil, corn oil, rape-seed oil, sunflower oil and peanut oil." Two problems arise:

Firstly, the source is not cited in the captions of Figures 2 and 3. This further raises the question, whether permission to reproduce these spectra/chromatograms was obtained. (Unless the authors obtained them themselves, in which case this issue is resolved.)

Secondly, there is no legend in Figures 2 and 3. In Figure 2, this makes it impossible to know which spectrum corresponds to Argan oil, corn oil, rape-seed oil, sunflower oil, and peanut oil (the discrimination between different oils being the very point of this review). In Figure 3, numbers designate the peaks observed in argan oil, but they are not explained. As a side note, the digital format of Figure 3 is not suitable (instead of crisp contrast, a grayish background surrounds text and other objects).

10- 224ff, 233ff: [18] and [35] use MIR, but the title and several passages of this section speak exclusively of N

We agree with the reviewer,. We removed (NIR) from the title. ‘’4.1.1. UV-Visible and Infrared spectroscopy’’ (See line: 229)

The original issue has been resolved.

However, Table 2 now differentiates "NIR" (near infrared) from "FTIR" (Fourier-transform infrared). FTIR can be both near and mid infrared. This implies that the study summarized as "NIR" was performed without FT, while not specifying in which range the FTIR studies were performed.

Please confirm (and clearly state in the caption) that the NIR was measured without FT, and state in which range the FTIR studies were performed.

11- 248"to assess the thermal stability..., to characterize oils and fats..., and to estimate quality related properties of Argan oil....". Which aspects (outside thermal stability and quality related properties) were characterized?

In Fang et al. (2013), (Characterization of oils and fats by 1H NMR and GC/MS fingerprinting: Classification, prediction and detection of adulteration), NMR and GC/MS fingerprints have been used in combination with chemometric tools to characterize different types of oils and fats such as sunflower, corn, coconut and chicken fats... The oils were distinguished and adulteration was detected. (See line: 274-275)
This was added to the text.

The issue has been resolved.

12- 263"...NMR could be as efficient as the GC technique in term of quantification of fatty acids..." In chromatography, separation efficiency has a specific meaning that does not translate to spectroscopy. What exactly is meant by "efficient" in this context? 

In this context, “efficient” means accurate, given that the results from both methods were comparable. This was clarified in the text.

The issue has been resolved.

13- Table S2: Remove unwanted line breaks and empty lines.

Unwanted line breaks were removed. (See Table S2)

The issue has been resolved.

14- Table 3:
-in the line of [52], it should be specified which wave length of IR was used in that study (i.e., MIR)
-in the line of [30], the authors specify "FT-MIR." In the rest of the table, the use of FT is not specified (even when it was used such as in [52])

Suggested changes were made. (See Table 3)

The last entry of Table 3 still does not specify the range (near or mid) of ATR-FTIR .

15- 335-341: If the phenolic acid content is ultimately responsible for the differentiation between pure Argan oil and vegetable oil adulterated Argan oil: Are there more or less phenolic acids in Argan oil?

Argan oil has a higher percentage of phenolic acids than the other vegetable oils used in this study. This was clarified in the text. “Argan oil has a higher amount than the other vegetable oils” (See lines 365-366)

It may be helpful to add this phrase to the existing clarification, i.e. "Argan oil has a higher amount of phenolic acids than the other vegetable oils". The issue is then resolved.

16- Table 4:
-The phase of the HPLC method [1] is further characterized as normal phase (NP-HPLC). Why is this only done for [1] but not for the other instances of HPLC? If this implies RP-HPLC in all other cases, please make this explicit.

As suggested by the reviewer, the chromatographic mode has been added to table 4. (See Table 4)

The issue has been resolved.

The abbreviations are useful, make the tables more concise, and for the largest part known to analytical chemists. Nevertheless, an abbreviation directory would make the manuscript more accessible to a wider readership (e.g., not everybody may be familiar with "GLC").

-Similarly, the column of the GC method [66] is further characterized as capillary column. But not all other GC methods used packed columns.

Thank you for this remark. In all studies, a capillary column was used.

The issue has been resolved.

Some redundancies can be eliminated (e.g., "Determination of...with PLS-DA" in the line of [38], since PLS-DA are already stated in the column for data analysis methods) and abbreviations (ELSD for evaporative light-scattering detector – especially since this abbreviation is used later in the manuscript; FD for fluorescence detection) may free up space.

As suggested by the reviewer, redundancies have been eliminated. (See Table 4)

The issue has been resolved.

It is difficult to delineate the different studies when they span more than one line. Top alignment or add delimiting lines may help.

We agree with the reviewer. To distinguish between the various studies, table 4 has undergone minor adjustments. (See Table 4)

The issue has been resolved.

17- 435-436: "...and the quantified chemicals are used in the data analysis." Does this mean quantitative data was required, while qualitative data by itself was insufficient? Please clarify.

Quantitative data are essential since they provide quantitative information on the various oils' component. However, a more complex matrix is produced when we quantify many substances simultaneously while using multiple samples (e.g., different type of oils). In this case, chemometric tools such as Cluster analysis, PCA, which Rezanková et al., 1999 utilized in their paper, or DA and OPLS-DA as in Miklavi et al., 2020, make it easier to analyze and extract relevant information from the data. This is what we call targeted analysis.

The issue has been resolved.

18- 487-491: The authors describe the need for greener chromatographic methods for Argan oil analysis. Previously, SFE was described as a favorable extraction method. This raises an interesting question: are there no reports of SFC methods for the characterization of Argan oil?

Greener and environmentally friendly chromatographic techniques are required for the analyses of Argan oil. However, to the best of our knowledge, no study has characterized and evaluated the quality of Argan oil using supercritical fluid chromatography.

Thank you for the clarification. The lack of SFC methods may be worth mentioning in the context of calling for environmentally-friendly analytical techniques for a sample type for which SFE methods exist (and therefore can be dissolved in scCO2).
